# Effect of Combined Treatment with Cinnamon Oil and *petit*-High Pressure CO_2_ against *Saccharomyces cerevisiae*

**DOI:** 10.3390/foods11213474

**Published:** 2022-11-02

**Authors:** Liyuan Niu, Jingfei Liu, Xinpei Wang, Zihao Wu, Qisen Xiang, Yanhong Bai

**Affiliations:** 1College of Food and Bioengineering, Zhengzhou University of Light Industry, Zhengzhou 450001, China; 2Henan Key Laboratory of Cold Chain Food Quality and Safety Control, Zhengzhou 450001, China; 3Collaborative Innovation Center of Food Production and Safety, Henan Province, Zhengzhou 450001, China

**Keywords:** cinnamon oil, *petit*-high pressure CO_2_, synergistic effect, *Saccharomyces cerevisiae*

## Abstract

This study investigated the effects of the combined treatment with cinnamon oil (CIN) and *petit*-high pressure CO_2_ (*p*-HPCO_2_) against *Saccharomyces cerevisiae*. The results showed that CIN and *p*-HPCO_2_ exhibited a synergistic antifungal effect against *S. cerevisiae*. After being treated by CIN at a final concentration of 0.02% and *p*-HPCO_2_ under 1.3 MPa at 25 °C for 2 h, the *S. cerevisiae* population decreased by 3.35 log_10_ CFU/mL, which was significantly (*p* < 0.05) higher than that of CIN (1.11 log_10_ CFU/mL) or *p*-HPCO_2_ (0.31 log_10_ CFU/mL). Through scanning electron microscopy, fluorescence staining, and other approaches, a disorder of the structure and function of the cell membrane was observed after the CIN + *p*-HPCO_2_ treatment, such as severe morphological changes, increased membrane permeability, decreased cell membrane potential, and loss of membrane integrity. CIN + *p*-HPCO_2_ also induced mitochondrial membrane depolarization in *S. cerevisiae* cells, which could be associated with the decrease in intracellular ATP observed in this study. Moreover, the expression of genes involved in ergosterol synthesis in *S. cerevisiae* was up-regulated after exposure to CIN + *p*-HPCO_2_, which might be an adaptive response to membrane damage. This work demonstrates the potential of CIN and *p*-HPCO_2_ in combination as an alternative pasteurization technique for use in the food industry.

## 1. Introduction

In the food industry, a variety of chemicals are usually used to inhibit pathogenic and spoilage microorganisms, such as sorbic acid and sodium benzoate [1]. However, owing to the serious health risks involved in the consumption of food products processed with synthetic additives and their immediate influences on the environment, demands for safe foods with the addition of natural preservatives have been growing in the last years [2,3]. Natural antimicrobials are bioactive substances derived from natural sources, such as essential oils (EOs) primarily from plants, lysozymes from animals, and bacteriocins from bacteria [3].

Plant EOs are volatile oily liquids extracted from a wide range of plants by extrusion, distillation, fermentation, and solvent extraction. As a type of natural antimicrobial, EOs extracted from plants (nutmeg, cinnamon, oregano, rosemary, clove, etc.) and their bioactive components have been included on the list of substrates that are generally recognized as safe (GRAS) by the United States Food and Drug Administration (FDA) [4]. The antimicrobial effect of EOs mainly depends on the active components, such as aldehydes, terpenes, and phenols [5]. These antimicrobial compounds can cause a disorder of the function and structure of cellular membranes and interfere with the cellular metabolic processes or inhibit the synthesis of nucleic acids and proteins [6,7,8], thereby leading to cell death. However, the dose of EOs required to exert a remarkable germicidal effect often exceeds the consumers’ acceptability because of the strong flavor [9,10]. Therefore, several studies have applied plant EOs in combination with other food processing and preservation approaches, such as pulsed magnetic fields, cold plasma, and thermo-ultrasound. A combined treatment can reduce the dose of individual antimicrobial agents and maintain food quality while ensuring microbial food safety. Sánchez-Rubio et al. (2018) reported the synergistic antifungal effect of thermo-ultrasound (24 kHz, 105 μm, 33.31 W/mL) and cinnamon leaf essential oil (650 ppm) against *Saccharomyces cerevisiae* in natural orange juice [11]. The combination of olive leaf extract and high-hydrostatic pressure (HHP) (400 MPa, 4 and 6 min) effectively maintained the physiochemical characteristics and microbial safety of Spanish-style table olive fermented with *S. cerevisiae* during storage [12]. Moreover, the combination of *Litseacubeba* oil (1.5 mg/mL) and pulsed magnetic fields (3 times under 8 T, 60 pulses) also exhibited a synergetic antimicrobial effect on *Escherichia coli* O157:H7 in vegetable juices, showing no influence on the sensory properties [13].

Cinnamon oil (CIN) is generally extracted from the bark and leaves of the medical and edible plant cinnamon. Due to its excellent antimicrobial, anti-inflammatory, and antioxidant effects [14], CIN appears very promising as a preservative in food storage. However, a high dose of CIN has a strong flavor and may have the potential for toxicity, which restricts its commercialization [15]. *petit*-high pressure CO_2_ (*p*-HPCO_2_) is a nonthermal pasteurization technique with many advantages, such as low price, ease of obtaining, and no residue [16]. The pressure used in the *p*-HPCO_2_ technology is usually no more than 1.30 MPa, which significantly reduces the equipment investment compared to high-pressure CO_2_ (generally > 7.38 MPa) and high-hydrostatic pressure (∼300–600 MPa) technologies. However, to achieve the desired antimicrobial activity, treatment with *p*-HPCO_2_ at room temperature takes a long time [16]. The previous study demonstrated that CIN and *p*-HPCO_2_, in combination, had synergistic antibacterial activity against *Salmonella typhimurium* [17]. The combined treatment could overcome the shortcomings of an individual treatment, which not only reduced the additive amount of CIN but also shortened the processing time and improved the germicidal efficiency. Food-associated yeasts that can survive under unfavorable conditions of high sugar, low water activity, and low pH are underestimated causes of foodborne disease [18]. However, the synergism between the antifungal activity of CIN and *p*-HPCO_2_ is still unclear.

*S. cerevisiae* is a common spoilage yeast typically found in osmophilic foods, such as fruit and vegetable juices [18]. The main objectives of this study were to investigate: (a) the inactivation effect on *S. cerevisiae* of CIN and *p*-HPCO_2_, individually or in combination; (b) the synergetic antifungal interaction between CIN and *p*-HPCO_2_; (c) their synergetic antifungal mechanism. This study provided a theoretical basis for the potential use of CIN combined with *p*-HPCO_2_ in food processing and preservation as an alternative technology.

## 2. Materials and Methods

### 2.1. Materials and Yeast Strain

The strain *S. cerevisiae* (ATCC204508) was used as the target microorganism in this study. A loopful of yeast cells, stored at −80 °C, were inoculated onto a Yeast Extract Peptone Dextrose medium (YPD, Qingdao Haibo Biotechnology, Qingdao, China) and cultivated at 25 °C for 48 h. A single colony was inoculated into 50 mL of fresh YPD medium and cultivated at 150 rpm in a shaker at 25 ℃ for 12 h. The yeast cells were washed twice with 25 mL of sterile saline solution (0.85% NaCl) and then resuspended in saline to obtain yeast suspensions with a final concentration of about 10^7^ CFU/mL by adjusting the optical density at 600 nm (OD_600_) to 1.0. CIN (ρ = 1.04 g/mL at 25 °C; GC-MS: 91.76% *trans*-cinnamaldehyde, 4.27% *trans*-cinnamic acid, nominal values) extracted by steam was purchased from Shanghai Yuanye Biotechnology Co., Ltd. (Shanghai, China). The CIN was stored in the dark at 4 °C before use.

### 2.2. The Minimum Inhibitory Concentration of CIN

The minimum inhibitory concentration (MIC) of CIN against *S. cerevisiae* was obtained using a broth microdilution method [17]. The MIC indicated the lowest concentration of CIN that completely inhibited microbial growth after incubation for 24 h. In brief, the stock solution of CIN was first dissolved in Tween 80 at a ratio of 1:1 (*v*/*v*) and then mixed with the sterile YPD medium to prepare the CIN solutions with concentrations of 0.0025–0.04% (*v*/*v*). To obtain a CIN solution at a concentration of 0.04%, 1 μL of dissolved CIN solution (50%) was added to 1200 μL of YPD medium. The other CIN solutions were then made with a 2-fold dilution method. Afterward, 100 µL of cell suspension in YPD and the equal CIN dilution were mixed in a 100-well microtiter plate. The final concentration of yeast cells in each well was about 10^6^ CFU/mL. Tween 80, with different concentrations corresponding to that in each work solution, was added to the yeast suspension to monitor its antifungal activity. The OD_600_ values of the *S. cerevisiae* cells were detected automatically every 2 h at 37 °C using a Microbiology Reader BioScreen C (BioScreen, Vantaa, Finland).

### 2.3. Treatment with CIN Combined with p-HPCO_2_

The OST-200ML HPCO_2_ system (Xi’an Oster Instrument Technology, Xi’an, China) was applied to inactivate *S. cerevisiae*. The pasteurization process, with CIN and *p*-HPCO_2_ and the *p*-HPCO_2_ setup, has been described in detail in our previous study [17]. The prepared yeast samples were transferred to the high-pressure vessel, which was placed in a thermostatic bath to control the process temperature. The samples were treated with CIN at different final concentrations (1/2×, 1×, or 2× MIC) and *p*-HPCO_2_ under 1.3 MPa at 25 °C for 2 h, individually or in combination. The number of *S. cerevisiae* surviving after treatment was determined by the plate count method. The data were expressed as log_10_ CFU/mL.

### 2.4. Synergism Assessment

The synergistic antimicrobial effect between the CIN and *p*-HPCO_2_ treatment was evaluated according to a previously reported method [19]. The experimentally observed log reduction values of *S. cerevisiae* after the combined treatment were compared to the expected log reduction values, which were calculated using the given equation:(1)Expected log reduction value= A+B−AB/100.
where *A* and *B* are the log reduction values of *S. cerevisiae* after each individual treatment. When the experimentally obtained yeast reduction is higher than expected, it indicates that the combined treatment with CIN and *p*-HPCO_2_ has a synergistic effect.

### 2.5. Field Emission-Scanning Electron Microscopy

The changes in the cellular morphology of *S. cerevisiae* were observed by field emission-scanning electron microscopy (FE-SEM) according to a previously reported method [20]. Briefly, the yeast suspensions were treated by individual and combined CIN (0.02%) and *p*-HPCO_2_ (1.3 MPa) at 25 °C for 2 h, respectively. The collected cells were fixed in 2.5% (*v*/*v*) glutaraldehyde at 4 °C for at least 4 h and were then successively dehydrated in a series of gradient alcohols (10–100%; *v*/*v*). Afterward, the yeast cells were placed on the silicon wafers, dried by an antosamdri-815 CO_2_ critical point dryer (Tousimis Research Corp., Rockville, MD, USA), and then sputter-coated with gold. The changes in the cellular morphology of *S. cerevisiae* were examined by an FE-SEM (Sigma 300, Carl Zeiss, Oberkochen, Germany) at an accelerating voltage of 3.0 kV.

### 2.6. Leakage of Intracellular DNA

The content of intracellular DNA leaked from the *S. cerevisiae* cells was measured to assess the membrane permeability. After being treated by the individual and combined CIN (0.02%) and *p*-HPCO_2_ (1.3 MPa) for 2 h at 25 °C, respectively, yeast suspensions were centrifuged at 4000× *g* for 15 min at 4 °C. The DNA contents in the supernatant solutions were quantified using a circulating DNA kit (TIANGEN Biotech Co., Ltd., Beijing, China) according to the instructions of the manufacturer.

### 2.7. Cellular Membrane Integrity

The membrane integrity of the *S. cerevisiae* cells was assessed by fluorescent staining with propidium iodide (PI) [21]. After being treated by the individual and combined CIN (0.02%) and *p*-HPCO_2_ (1.3 MPa) for 2 h at 25 °C, respectively, the yeast samples were mixed with the PI solution at a final concentration of 3 µmol/L and then incubated at 37 °C in the dark for 15 min. After that, the yeast cells were rinsed twice with a sterile saline solution and resuspended in 1 mL of the saline. The fluorescence intensity was evaluated by the Tecan Spark^®^ multimode microplate reader (Tecan Group Ltd., Männedorf, Switzerland) at an excitation (E_x_)/emission (E_m_) wavelength of 485/635 nm. The relative fluorescence intensity of PI was calculated as the ratio of the fluorescence intensity of the treated yeast suspensions to that of the untreated yeast suspension, which represented the membrane permeability of the yeast cells. A fluorescence microscope (Nikon Eclipse 80i; Nikon, Tokyo, Japan) was also applied to examine the cellular uptake of PI.

### 2.8. Cellular Membrane Potential

The cellular membrane potential was evaluated by the method of DiBAC_4_(3) staining [20]. After each treatment, the mixtures of yeast samples and DiBAC_4_(3) probe, at a final concentration of 1.0 µg/mL, were incubated at 37 °C for 30 min in the dark. The samples were then washed and resuspended in a sterile saline solution. The fluorescence intensity was evaluated by the Tecan Spark^®^ multimode microplate reader (Tecan Group Ltd., Männedorf, Switzerland) at E_x_/E_m_ of 488/525 nm. The relative fluorescence intensity of DiBAC_4_(3) represented the membrane potential of the yeast cells and was calculated as the ratio of the fluorescence intensity of the treated yeast suspensions to that of the untreated samples.

### 2.9. Expression of the Genes Related to Ergosterol Biosynthesis

The *S. cerevisiae* cells were treated by CIN (0.02%), *p*-HPCO_2_ (1.3 MPa), and CIN + *p*-HPCO_2_ for 2 h at 25 °C, respectively. A fungal total RNA isolation kit (Sangon Biotech, Co., Ltd. Shanghai, China) was used to extract the total RNA in yeast. Then, the cDNA was synthesized from the total RNA using the ReverTra Ace^®^ qPCR RT Master Mix (TOYOBO, Osaka, Japan). A PCR was carried out with the cDNA as a template. Table 1 lists the primer pairs used in this work. The quantitative real-time PCR was conducted using SYBR Green I (GenStar, Beijing, China) on a Roche LightCycler^®^ 480 Instrument II (Roche Applied Science, Mannheim, Germany). The PCR condition included the initial denaturation at 95 °C for 2 min followed by 45 cycles of 95 °C for 15 s and 60 °C for 30 s; and a final extension at 72 °C for 20 s. The relative gene expression was expressed as the fold change calculated by the 2^-ΔΔCT^ normalization of the *ACT1* gene (actin).

### 2.10. Mitochondrial Membrane Potential

A fluorescent probe JC-1 was used to measure the changes in the mitochondrial membrane potential (∆ψm) of *S. cerevisiae* [21]. After exposure to CIN (0.02%), *p*-HPCO_2_ (1.3 MPa), and CIN + *p*-HPCO_2_ for 2 h at 25 °C, respectively, the yeast cells were harvested by centrifugation and resuspended in 500 µL of YPD medium. Then, the cell suspension and 500 µL of JC-1 staining solution were mixed and cultivated at 37 °C for 20 min in the dark. After incubation, the cells were washed twice with a sterile saline solution and resuspended in the saline. The fluorescence intensity of the JC-1 dye in monomeric forms and J-aggregates was detected using the Tecan Spark^®^ multimode microplate reader (Tecan Group Ltd., Männedorf, Switzerland) at E_x_/E_m_ of 525/590 nm and 490/530 nm, respectively. The mitochondrial membrane potential was expressed as the ratio of the ∆ψm value of the treated yeast suspensions to that of the untreated samples, where ∆ψm indicated the ratio of red-to-green fluorescence intensity. A fluorescence microscope (Nikon Eclipse 80i; Nikon, Tokyo, Japan) was applied to obtain the fluorescent images of the yeast cells stained with JC-1.

### 2.11. Intracellular ATP Content

After being treated with CIN (0.02%), *p*-HPCO_2_ (1.3 MPa), or CIN + *p*-HPCO_2_ for 2 h at 25 °C, *S. cerevisiae* cells were washed twice with 1 mL of sterile saline solution and resuspended in the cell lysis solution. The mixtures were cultured for 10 min at 25 °C and boiled for 10 min at 100 °C to fully release intracellular ATP. Afterward, the samples were immediately placed on ice to cool down. The supernatants were collected to measure the content of intracellular ATP according to the instructions of an ATP assay kit (Beyotime Biotechnology, Shanghai, China) [17]. The Tecan Spark^®^ multimode microplate reader (Tecan Group Ltd., Männedorf, Switzerland) was used to detect the luminescence intensity.

### 2.12. Statistical Analysis

All experiments were carried out in triplicate. We used one-way analysis of variance (ANOVA) and Ducan’s multiple range tests to analyze the obtained data using SPSS Statistics 22.0 software (IBM SPSS Statistics, New York, NY, USA). The results were expressed as the mean ± standard deviation (SD). A *p* < 0.05 was considered a statistically significant difference.

## 3. Results and Discussion

### 3.1. The MIC of CIN

Figure 1 shows the growth curves of *S. cerevisiae* cultivated in the presence of CIN at different concentrations. When the concentration of the CIN was 0.005%, an apparent growth delay occurred in the *S. cerevisiae* culture compared to the untreated samples. CIN, at a concentration of 0.0025%, did not exhibit any antifungal effect. The yeast growth was completely inhibited when the concentration of CIN was 0.01%. The observations demonstrated that the CIN used in this work had a great inhibitory effect against *S. cerevisiae*, with a MIC value of 0.01%. However, Le et al. [22] and Denkova-Kostova et al. [23] reported that the MIC value of CIN against *S. cerevisiae* was 0.0006% (*v*/*v*) and 20% (*w*/*v*), respectively. These differences may result from the different harvest seasons, provenances, and plant parts, where the CIN was extracted, which can further influence the chemical composition and antimicrobial activity of CIN. Our previous study has reported that *trans*-cinnamaldehyde (91.76%) was the major component of CIN used in this work, which was likely to contribute largely to the antifungal activity [17].

### 3.2. Antifungal Analysis of CIN + p-HPCO_2_

Figure 2 shows the antifungal effects of CIN, *p*-HPCO_2_, and their combination. At an initial microbial load of 7.18 log_10_ CFU/mL, the treatment with *p*-HPCO_2_ alone under 1.3 MPa or CIN alone at different concentrations (0.005%, 0.01%, 0.02%) at 25 °C for 2 h only, decreased the *S. cerevisiae* cells by less than 1.12 log_10_ CFU/mL. However, the yeast population decreased by 1.55–3.35 log_10_ CFU/mL after exposure to the CIN combined with *p*-HPCO_2_. The results demonstrated that the fungicidal effect of CIN + *p*-HPCO_2_ on *S. cerevisiae* was significantly greater than that of the CIN or *p*-HPCO_2_ alone treatment (*p* < 0.05), and it was enhanced with the increase in the CIN concentration. Table 2 shows the results of the synergistic antifungal activity of CIN and *p*-HPCO_2_. The ratio of the experimentally obtained yeast reduction to the expected reduction after combined treatment was 1.97, 3.24, and 2.36, respectively, when the concentration of applied CIN was 0.005%, 0.01%, and 0.02%, indicating great synergism between the antifungal effect of CIN and *p*-HPCO_2_ against *S. cerevisiae*. Our previous study also observed a synergistic bactericidal effect of CIN and *p*-HPCO_2_ against *S. typhimurium* [17]. These findings demonstrated that this combined method could be effective for the inactivation of both bacteria and fungi, showing its potential as an alternative nonthermal technology for food pasteurization.

### 3.3. Cell Morphological Characterization

The changes in the cellular morphology of *S. cerevisiae* after each treatment were visually observed by the FE-SEM. As depicted in Figure 3A, the untreated *S. cerevisiae* cells were intact and regularly ellipsoidal or spherical, and there was no obvious damage to the cell surface. However, when treated with CIN or *p*-HPCO_2_ alone, the surface of the yeast cells became wrinkled and dimpled (Figure 3B,C). The application of CIN and *p*-HPCO_2_ simultaneously induced more serious changes in the cellular morphology of *S. cerevisiae* (Figure 3D); the shrinkage was aggravated, and the dimples were increased on the cell surface. The damage that occurred to the yeast cells could be accompanied by the leakage of inner-cell materials. The severity of cell-surface deformation was consistent with the inactivation results obtained by plate colony counting (Figure 2).

On the one hand, a major compound of CIN, *trans*-cinnamaldehyde, was reported to be an active cell-wall antifungal agent because it was able to inhibit the enzymes involved in the biosynthesis of the cell wall and its components of *S. cerevisiae* (such as chitin synthase and *β*-(1,3)-glucan synthase), which was unfavorable for the maintenance of cell wall integrity [24]. The disturbance of the cell membrane of *S. cerevisiae*, induced by *p*-HPCO_2_ stress, was also observed in a previous study [20]. It has been theoretically confirmed that the CO_2_ molecule has a high affinity for cell membranes [25]. The damage to the cell wall that CIN induces could contribute to the accumulation of lipophilic CO_2_ and CIN or its components in the phospholipid bilayer, furthering the disorder/alteration of the cell membrane structure and function. On the other hand, a previous study has reported that pressurized CO_2_ could decrease the pH of the suspending solution because of the increased solubility of CO_2_ under pressure [25]. A low pH is helpful for the EOs to dissolve in the phospholipids layer of the cell membrane because the hydrophobicity of the EOs increases as the pH decreases [26]. Consequently, the combination of CIN and *p*-HPCO_2_ was able to aggravate the damage to the cell envelope, promoting yeast inactivation.

### 3.4. Cell Membrane Permeability

Cell morphological changes, presented in the FE-SEM images (Figure 3), could lead to the release of cytosolic components. Therefore, the leakage of intracellular DNA from *S. cerevisiae* was investigated after treatment with CIN, *p*-HPCO_2_, and their combination, respectively. As presented in Figure 4, the amount of extracellular DNA after exposure to CIN and *p*-HPCO_2_ increased by 5.60 and 2.53 mg/mL, respectively, compared to the control (37.66 mg/mL), while for the combined treatment, the content was increased by 9.20 mg/mL, which is significantly higher than that for each individual treatment (*p* < 0.05). The results indicated that the combination of CIN + *p*-HPCO_2_ promoted the leakage of intracellular contents from *S. cerevisiae* cells. The findings were consistent with a study on *S. cerevisiae*, reported by Smid et al. (1996), that *trans*-cinnamaldehyde led to a partial collapse of the plasma membrane and the leakage of cellular enzymes and metabolites [27]. Moreover, the cell membrane permeability of *S. cerevisiae* could also interfere with pressurized CO_2_ [25]. Membrane permeability ensures that substances can selectively pass through the membrane, which is very crucial for the normal physiochemical functions of cells [28]. In this work, the antifungal action of CIN + *p*-HPCO_2_ was probably related to its capability to enhance the membrane permeability of *S. cerevisiae*, thereby resulting in the leakage of intracellular contents.

### 3.5. Plasma Membrane Integrity

A nucleotide-binding dye, PI, is commonly used to estimate cell membrane integrity. It can enter through the destructed membranes and bind to intracellular nucleic acids, emitting a red fluorescence [17]. Figure 5b presents the relative fluorescence intensity of the PI in *S. cerevisiae* with individual and combined CIN and *p*-HPCO_2_ treatments. The fluorescence intensity of the PI probe in the untreated yeast suspension was considered 100%. With the treatment of CIN + *p*-HPCO_2_, the relative PI uptake was increased by 2.19-fold (*p* < 0.05), while for the individual treatment, it was not significantly different from the untreated samples (*p* > 0.05). The PI uptake was also observed using a fluorescence microscope. As depicted in Figure 5a, the PI-positive *S. cerevisiae* cells, with or without individual treatment, were hardly found, while the number of yeast cells with a red fluorescence remarkably increased after the combined treatment. The fluorescent images coincided with the obtained relative value of the PI uptake. It has been reported that *trans*-cinnamaldehyde or CIN and pressurized CO_2_ act on the plasma membrane of *S. cerevisiae* [25,27]. In this study, although the CIN or *p*-HPCO_2_ alone treatment did not remarkably affect the membrane permeability, membrane potential, and integrity of *S. cerevisiae*, the stability of the cell membrane could be weakened. When treated with CIN + *p*-HPCO_2_, damage to the plasma membrane occurred.

### 3.6. Cell Membrane Potential

Membrane potential alters through the changes in extracellular ion concentration, which plays a major role in the processes associated with the external stimulation of cells, such as signal transduction and ion transport across the membrane [29]. Changes in membrane potential involve either depolarization or hyperpolarization, and the former implies a decrease in cell activity but not cell death [29]. DiBAC_4_(3) dye can only enter depolarized cells and bind intracellular proteins, exhibiting enhanced fluorescence. The effects of individual and combination treatments on the DiBAC_4_(3) accumulation in *S. cerevisiae* cells are exhibited in Figure 6. After being treated by CIN, *p*-HPCO_2_, and CIN + *p*-HPCO_2_, the DiBAC_4_(3) accumulation in yeast cells was increased by 3.62-, 0.56- and 10.67-fold, respectively, compared to the control (100%). The results revealed that the combined treatment with CIN and *p*-HPCO_2_ induced more severe depolarization of the cell membrane than either the CIN or *p*-HPCO_2_ treatment alone, which confirmed the destruction of the cytoplasm membrane of *S. cerevisiae*. It coincided with previous studies [17,21] that there was a good correlation between cell viability and membrane damage.

The cell plasma membrane is a crucial barrier of protection from environmental stresses [28]. Our findings suggested that the cell membrane was a primary target of CIN + *p*-HPCO_2_ action on *S. cerevisiae*, which might be related to the lipophilicity of CIN or its components and CO_2_. These lipophilic substances could accumulate in the plasma membrane or penetrate yeast cells, causing serious alterations in the membrane structure and function, disturbing cellular homeostasis, and thus leading to a decline in cell activity.

### 3.7. Expression of Genes Related to Ergosterol Biosynthesis

The results mentioned above have found that CIN and *p*-HPCO_2_, in combination, affected the function and structure of the cell membranes of *S. cerevisiae* ( Figure 3, Figure 4, Figure 5 and Figure 6). In unicellular eukaryotic organisms, ergosterol is the most important constituent of the yeast cell membrane, which is essential to the survival of yeast strains under stress conditions because it is related to plasma membrane permeability and membrane-bound enzyme activity [30]. Yeast cells with deletions in genes related to ergosterol biosynthesis frequently show increased membrane permeability. Therefore, ergosterol is usually considered an effective target of antifungals and plays a crucial role in antimicrobial resistance [31].

To understand the effect of CIN combined with the *p*-HPCO_2_ treatment on the ergosterol in *S. cerevisiae*, the transcriptional levels of the *ERG10, HMG2*, *ERG8*, and *ERG6* genes involved in its biosynthesis pathway were investigated in this study (Figure 7). The *ERG10* gene encodes the enzyme acetyl-coenzyme A (CoA) acetyltransferase (ACAT) in *S. cerevisiae*. ACAT is the first enzyme that participates in the biosynthesis pathway of ergosterol and is reported to be essential for cell growth [32]. Mevalonate, the precursor of ergosterol, is critical for cell viability, the synthesis of which is catalyzed by the 3-hydroxy-3-methylglutaryl coenzyme A (HMG-CoA) reductase, encoded by the *HMG2* gene [30]. Moreover, the *ERG8* gene encodes phosphomevalonate kinase (PMK), which catalyzes an essential step in the mevalonate pathway [33]. These three genes, *ERG10*, *HMG2*, and *ERG8*, were found to be up-regulated by 2.68-, 2.22-, and 1.85-fold, respectively, after the CIN + *p*-HPCO_2_ treatment, which was higher than each individual treatment, while the *ERG6* gene expression was not significantly affected. This suggested that the combination treatment could induce the biosynthesis of ergosterol.

Similar results have been reported in previous studies, that the mRNA levels of a set of genes related to the ergosterol biosynthesis in *S. cerevisiae* were observed as up-regulated after exposure to d-limonene [30] and *α*-terpinene [34], derived from aromatic plants. The enhancement of ergosterol synthesis contributes to maintaining the normal function and structure of the cell plasma membrane to improve the tolerance of *S. cerevisiae* against environmental stress. Since CIN or its components and CO_2_ are lipophilic, a combined treatment with them could cause an occurrence of stress in the cellular lipid environment, which would probably induce an adaptive response to protect the cell wall-membrane system from stress damage [34].

### 3.8. Mitochondrial Membrane Potential

The mitochondrial stain, JC-1, is a fluorescent and membrane-permeant carbocyanine dye. This probe can accumulate as aggregates form in the membrane of healthy mitochondria with high ∆ψm, exhibiting intense red fluorescence, while in the monomeric form, in the mitochondrial membrane with diminished ∆ψm, it emits green fluorescence [21]. The ratio of red-to-green fluorescence intensity indicates the occurrence of mitochondrial membrane depolarization. After the combined treatment, most of the yeast cells emitted an intense green fluorescence (Figure 8a). The red/green fluorescence ratio of yeast cells with the CIN + *p*-HPCO_2_ treatment relative to the control was decreased to 48.59%, significantly higher than that of the CIN or *p*-HPCO_2_ alone treatment (70.02% and 84.10%, respectively) (Figure 8b). The results suggested the destruction of the mitochondrial membrane potential of *S. cerevisiae*. Mitochondria are the energy-producing centers in the eukaryotic cells, and their dysfunctions are the landmark event in the process of apoptosis. The depolarization of the mitochondrial membrane can hinder the synthesis and ∆ψm-dependent transport of ATP, thereby disturbing cellular metabolism [35,36].

### 3.9. Intracellular ATP

As is well known, the major function of mitochondria is to synthesize ATP. Considering the alteration of the mitochondrial membrane potential in *S. cerevisiae*, the intracellular ATP content was analyzed according to the ATP luminescence assay. As presented in Figure 9, the intensity of the luminescence of ATP in the cell interior was remarkably decreased by 67.34% and 90.37% after being treated by CIN alone or combined with *p*-HPCO_2_, respectively. There was no obvious change in the ATP content after the *p*-HPCO_2_ treatment. The observations were consistent with previous studies that the intracellular ATP content was closely correlated to the viable cell population [17,37]. As observed above, severe damage to the cell envelope of *S. cerevisiae* resulted from the combination of CIN and *p*-HPCO_2_ contributing to the CIN or *trans*-cinnamaldehyde and CO_2_ molecules accumulating into the cell interior. On the one hand, CIN or *trans*-cinnamaldehyde could reduce the F_1_F_0_-ATPase activity, thereby hindering ATP synthesis in microbial cells [38]. On the other hand, adaptive responses of cells to stress could increase the metabolic rate and ATP consumption [25,39]. For example, CO_2_ in the cells could dissociate into various ionic species, such as hydrogen (H^+^), bicarbonate (HCO_3_^−^), and carbonate (CO_3_^2−^). The ATP consumption in *S. cerevisiae* cells could be increased by excreting the excess protons by the H^+^-ATPase system or vacuolar proton-translocating ATPases (V-ATPase) to maintain cellular pH homeostasis [25]. Moreover, the leakage of ATP from the cellular interior due to the destruction of membrane integrity may also be one reason for the decrease in intracellular ATP. In conclusion, CIN, in combination with *p*-HPCO_2_, enhanced the inhibitory effect on the energy metabolism in *S. cerevisiae* cells.

### 3.10. Synergistic Antifungal Mechanism of CIN + p-HPCO_2_

Based on the findings presented above, the synergistic antifungal mechanism of CIN and *p*-HPCO_2_ against *S. cerevisiae* was proposed (Figure 10). The treatment with CIN + *p*-HPCO_2_ destructed the function and structure of the cell membrane first, exhibiting increased membrane permeability and accelerated membrane depolarization, and even a loss of membrane integrity. This could affect the selective transport of substances across the cell membrane and signal transduction and cause the leakage of intracellular material, such as DNA. Moreover, CIN combined with *p*-HPCO_2_ also led to the destruction of the mitochondrial membranes in yeast cells, which could be related to the decreased intracellular ATP, thereby disturbing energy metabolism. In conclusion, CIN combined with *p*-HPCO_2_ induced cell membrane destruction, mitochondrial membrane depolarization, and intracellular ATP depletion of *S. cerevisiae* cells, which could eventually be responsible for yeast death.

## 4. Conclusions

The use of CIN in combination with *p*-HPCO_2_ was found to be effective for the inactivation of *S. cerevisiae*. A synergistic antifungal effect by CIN and *p*-HPCO_2_ was observed. Treatment with CIN, at a concentration of 0.02%, combined with *p*-HPCO_2_ at 25 °C for 2 h, achieved a 3.35-log reduction of *S. cerevisiae*, which is significantly greater than the individual treatment. The combined treatment caused severe damage to the *S. cerevisiae* cells, including functional and structural disturbances of the cell membrane, DNA leakage from the cell interior, mitochondrial membrane depolarization, and a decrease in intracellular ATP. At the transcriptional level, the expression of the genes associated with ergosterol biosynthesis in *S. cerevisiae* was also found to be up-regulated by CIN + *p*-HPCO_2_, conferring cellular resistance to membrane damage. The antifungal synergistic mechanism of CIN and *p*-HPCO_2_ should be further explored by focusing on other cellular targets and using multi-omics technologies. This study demonstrates that CIN combined with *p*-HPCO_2_ has great potential as an alternative method for food pasteurization. Additionally, because of the more limited diffusion of CO_2_ into the solid matrices, this novel technology is most suitable for the processing and preservation of liquid foods. It is noted that when this combined technology is applied in the food system, the technical parameters should be further explored and optimized according to the actual situation, such as the temperature, acidity, and concentration of CIN. Our previous study has proved that a proper increase in the process temperature could enhance the inactivation efficiency and shorten the treatment time [17]. Moreover, the influences of CIN in combination with *p*-HPCO_2_ on food products’ microbial safety and sensory properties should also be verified.

## Figures and Tables

**Figure 1 foods-11-03474-f001:**
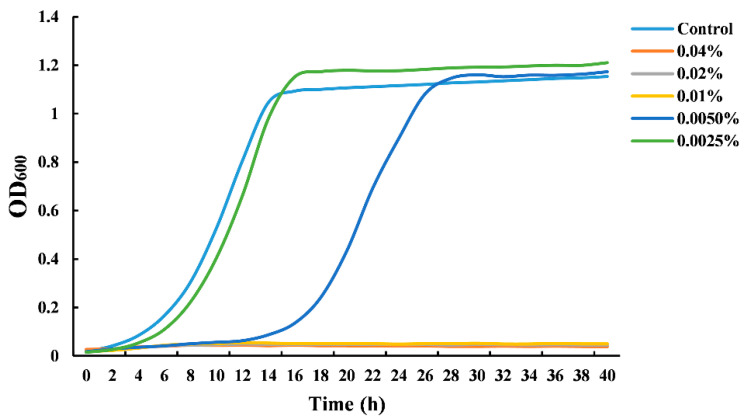
Growth curves of *Saccharomyces cerevisiae* grown in the presence of cinnamon oil (CIN).

**Figure 2 foods-11-03474-f002:**
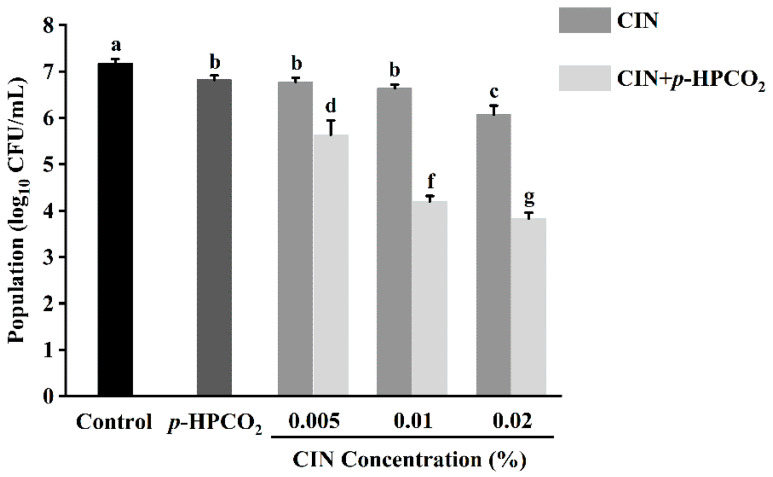
Survival of *Saccharomyces cerevisiae* after treatment with cinnamon oil (CIN), *petit*-high pressure CO_2_ (*p*-HPCO_2_), and CIN + *p*-HPCO_2_ under 1.3 MPa for 2 h at 25 °C. Different letters above bars mean significant differences (*p* < 0.05).

**Figure 3 foods-11-03474-f003:**
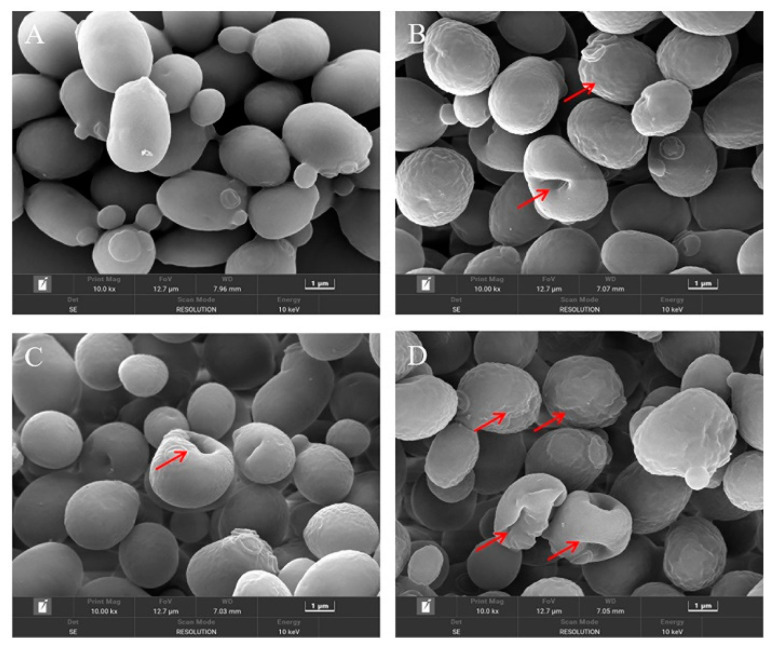
Field emission-scanning electron microscopy images of *Saccharomyces cerevisiae* cells after individual and combined treatments with cinnamon oil (CIN) (0.02%) and *petit*-high pressure CO_2_ (*p*-HPCO_2_) (1.3 MPa) for 2 h at 25 °C. (**A**) control; (**B**) CIN; (**C**) *p*-HPCO_2_; (**D**) CIN + *p*-HPCO_2_. The red arrows indicate the representative changes in cellular morphology.

**Figure 4 foods-11-03474-f004:**
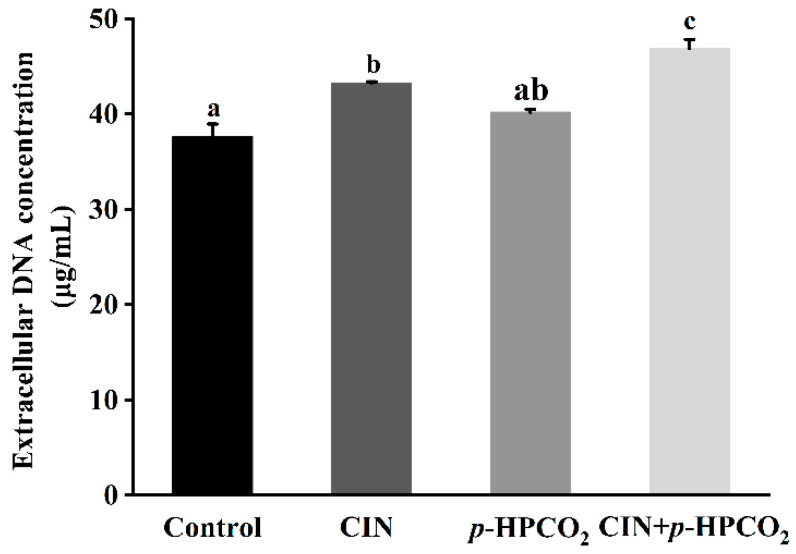
Extracellular DNA content of *Saccharomyces cerevisiae* after individual and combined treatments with cinnamon oil (CIN) (0.02%) and *petit*-high pressure CO_2_ (*p*-HPCO_2_) (1.3 MPa) for 2 h at 25 °C. Different letters above bars indicate significant differences (*p* < 0.05).

**Figure 5 foods-11-03474-f005:**
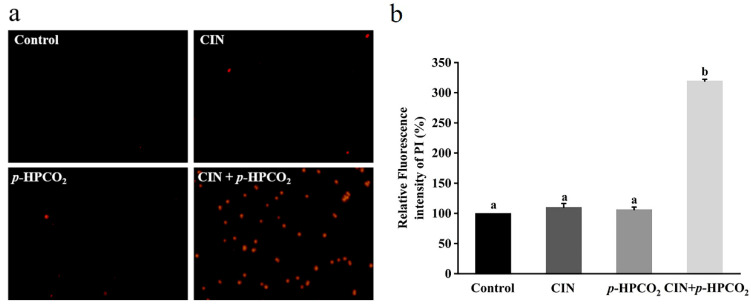
Fluorescence microscopic images of *Saccharomyces cerevisiae* cells stained with propidium iodide (PI) (×400) (**a**) and relative fluorescence intensity of PI in *S. cerevisiae* (**b**). *S. cerevisiae cells* were treated with individual and combined treatments with cinnamon oil (CIN) (0.02%) and *petit*-high pressure CO_2_ (*p*-HPCO_2_) (1.3 MPa) for 2 h at 25 °C. Different letters above bars indicate significant differences (*p* < 0.05).

**Figure 6 foods-11-03474-f006:**
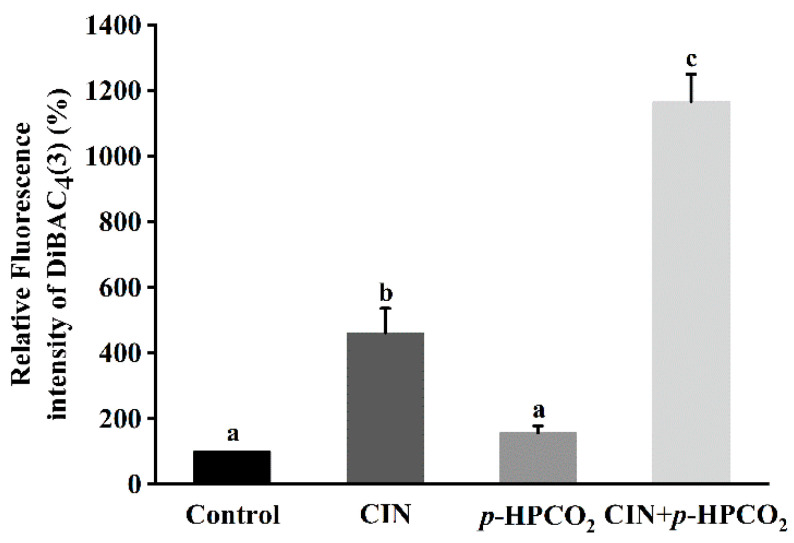
Relative fluorescence intensity of DiBAC_4_(3) in *Saccharomyces cerevisiae* after individual and combined treatments with cinnamon oil (CIN) (0.02%) and *petit*-high pressure CO_2_ (*p*-HPCO_2_) (1.3 MPa) for 2 h at 25 °C. Different letters above bars indicate significant differences (*p* < 0.05).

**Figure 7 foods-11-03474-f007:**
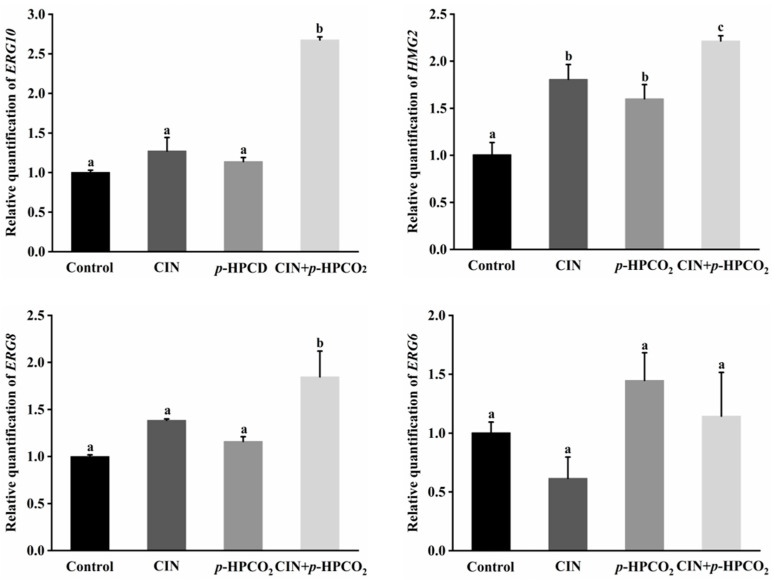
Relative expression of genes involved in ergosterol biosynthesis in *Saccharomyces cerevisiae* treated with individual and combined cinnamon oil (CIN) (0.02%) and *petit*-high pressure CO_2_ (*p*-HPCO_2_) (1.3 MPa) for 2 h at 25 °C. The transcriptional levels are expressed as fold increase or decrease relative to that of the control. The housekeeping gene *ACT1* was used as an internal standard. Different letters above bars indicate significant differences (*p* < 0.05).

**Figure 8 foods-11-03474-f008:**
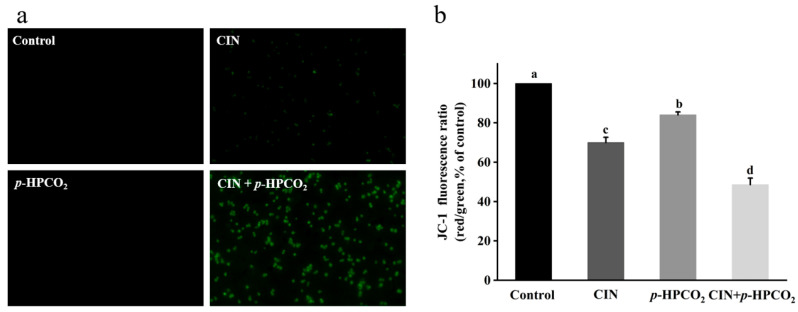
Fluorescent microscopic images of *Saccharomyces cerevisiae* cells stained with JC-1 (×400) (**a**) and relative red/green fluorescence intensity of JC-1 in *S. cerevisiae* (**b**). *S. cerevisiae* cells were treated with individual and combined cinnamon oil (CIN) (0.02%) and *petit*-high pressure CO_2_ (*p*-HPCO_2_) (1.3 MPa) for 2 h at 25 °C, respectively. Different letters above bars mean significant differences (*p* < 0.05).

**Figure 9 foods-11-03474-f009:**
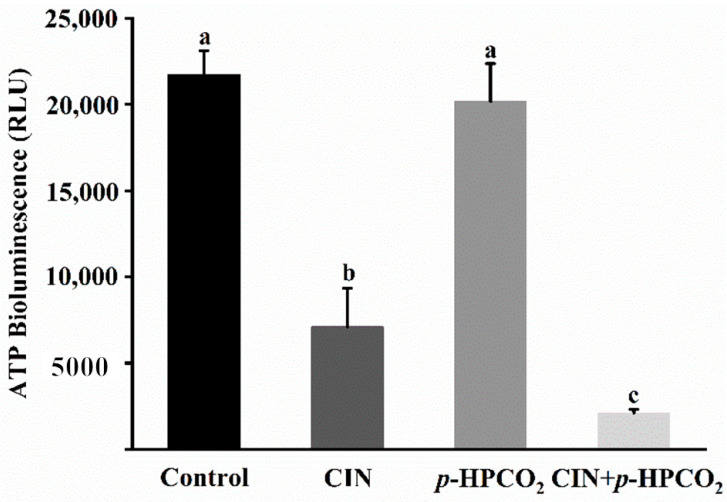
Intracellular ATP content in *Saccharomyces cerevisiae* treated with individual and combined cinnamon oil (CIN) (0.02%) and *petit*-high pressure CO_2_ (*p*-HPCO_2_) (1.3 MPa) for 2 h at 25 °C. Different letters above bars mean significant differences (*p* < 0.05).

**Figure 10 foods-11-03474-f010:**
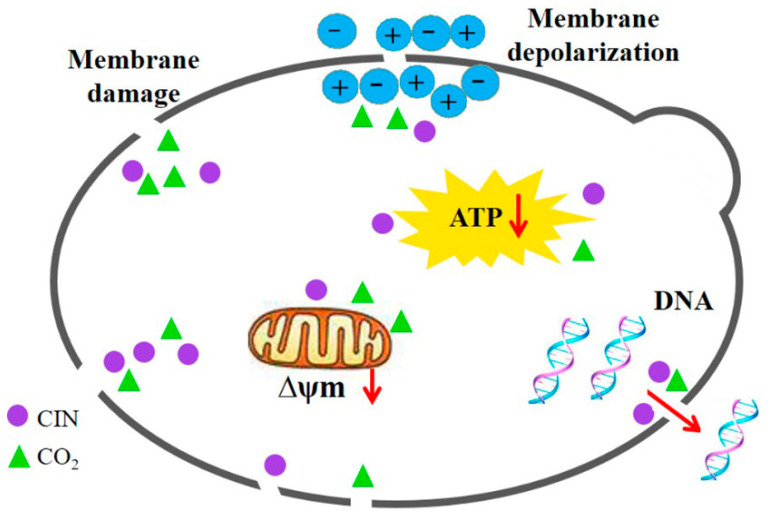
Possible synergistic antifungal mechanism of cinnamon oil (CIN) and *petit*-high pressure CO_2_ against *Saccharomyces cerevisiae*.

**Table 1 foods-11-03474-t001:** Primer pairs used for quantitative real-time PCR.

Gene	Primer Sequences (5′-3′)		Amplicon Size (bp)
*ACT1*	F	ACCGCTGCTCAATCTTCTTC	164
	R	ATGATGGAGTTGTAAGTAGTTTGG	
*HMG2*	F	GGTTGGGAAGATATGGAAGTTG	131
	R	ACGACATCACCAGGAATAGTAG	
*ERG6*	F	AAGACCTGGCGGACAATGATG	199
	R	AGAGCAGCAGTAACTTCCTTGG	
*ERG8*	F	AGTGGCTTCATTCCTGTTTCG	177
	R	TTCGGTAACGCTATCCTCCTG	
*ERG10*	F	TCCGCTATGAAGGCAATC	163
	R	CGACACCATCAACAAGAAC	

**Table 2 foods-11-03474-t002:** Synergism between antifungal effect against *Saccharomyces cerevisiae* of cinnamon oil (CIN) and *petit*-high pressure CO_2_ (*p*-HPCO_2_) under 1.3 MPa for 2 h at 25 °C.

	Observed Log Reduction	Expected Log Reduction	Ratio	Synergism
0.005%				
*p*-HPCO_2_	0.38			
CIN	0.40			
*p*-HPCO_2_ + CIN	1.54	0.78	1.97	Yes
0.01%				
*p*-HPCO_2_	0.38			
CIN	0.54			
*p*-HPCO_2_ + CIN	2.97	0.92	3.24	Yes
0.02%				
*p*-HPCO_2_	0.31			
CIN	1.11			
*p*-HPCO_2_ + CIN	3.34	1.42	2.36	Yes

## Data Availability

Data is contained within the article.

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
