# Peer review of "Effect of Combined Treatment with Cinnamon Oil and petit-High Pressure CO2 against Saccharomyces cerevisiae"

_foods, 2022, doi:10.3390/foods11213474_

Round 1
Reviewer 1 Report
The synergetic effect of cinnamon oil (CIN) combined with petit-12 high pressure CO2 (p-HPCO2) against Saccharomyces cerevisiae was reported. The study is interesting, and the results could interest the readership of Foods and pomologist. The authors need to revise their script by clarifying some sentences and improving others. Below are some points to address and help improve the manuscript
Abstract lacks methodology. Please add some methods. Also correct the grammar in the abstract
Line 29 Chemicals such as???
Line 30-32 do you have reference to buttress the statement?
Line 81 “107 CFU/mL” how was this done?
Line 42 “disorder the function” rephrase please
Line 45 “Exhibit” exert???
Line 46-50 please improve
Line 50-57 why no reference yeast instead of bacteria? These are two difference organisms hence react differently with treatments.
Line 59-60 please wording
LINE 76-77 quantity inoculated
Line 79 was with what? Acid, water, base and what volume used
Line 89 volume of ypd used ?
Line 92 was it an estimation or how was the final concentration calculated?
Line 95 (BioScreen, Finland) please include city? And do same to all equipment used
Line 181 wash with what and volume used?
Line 182 “boiled for 10 min at 100°C” why did you boil it?
Reviewer 2 Report
Manuscript investigates the demonstrates the potential of cinnamon oil and petit-high pressure CO2 in combination as an alternative pasteurization technique for use in the food industry against Saccharomyces cerevisiae. The manuscript is well written and demonstrates an alternative to pasteurization.
Some revisions should be done to improve the quality of the manuscript.
The aim of the work must be rewritten by clearly stating aim.
The p-HPCO2 technique should be explained deeply in the introduction part.
For what type of industry could the findings found in this manuscript be used?
More discussion of the results compared to the literature is necessary.
Maybe, in the introduction part you should write some aspect related to other essential oils, or even in the discussion part.
Why do not study the effect of this treatments on the phenol profile? For example, in this manuscript the effect of phenols was evaluated after HHP. Maybe, you should discuss it.
How the essential oil influence on the samples?
Why do not do the sensory analysis after the best treatment studied?
How do you mask the aroma applied with the oil added?
I would recommend discussing the introduction a little more with other works in the bibliography. For example, with the effect of Saccharomyces cerevisiae after a high hydrostatic pressure treatment.
Martín-Vertedor, D., Schaide, T., Boselli, E., Martínez, M., García-Parra, J., & Pérez-Nevado, F. (2022). Effect of High Hydrostatic Pressure in the Storage of Spanish-Style Table Olive Fermented with Olive Leaf Extract and Saccharomyces cerevisiae. Molecules, 27(6), 2028.
The grammar needs a thorough check. Is the whole text in the past simple tense or the present simple tense?
How do you obtain the essential oil? It should be explained in the material and method.
You should clarify the design of the experiments done. Maybe, I recommend you to make a diagram of the experiment graphically. This will help to understand the design.
The high-pressure equipment should be detailed in material and method. Why did you apply this pressure? During 2 hours? It seems that it is too aggressive a treatment. If such high-pressure treatments are applied, it implies a very high energy cost and, as expected, the microbial inhibition results obtained were successful. Although it is true that the pressure used was not very high, in the literature I have mentioned, high-pressure treatments were far superior, although with much shorter cycle times. Thus, you have to discuss this issue.
How to ensure that the treatment temperature was constant at 25ºC.
How many repetitions did you use?
What about the statistical analysis?
Maybe, you can rewritten the conclusion part according to the previous comments.
Round 2
Reviewer 2 Report
The authors have made a number of changes to the text that significantly improve the quality of the publication. The language used in the article has been improved. Thoughts and facts were exposed clearly. The well-written manuscript about has a significant relevance to food science research field. In the line 57-58, the reference introduce is not well writing according to Molecules guide author. Therefore, I believe that all necessary corrections have been made by the authors and the article should be considered for publication in Foods. I have no comments on the current content of the article. My sincere congratulations to the authors for their efforts and commitment with the scientific community.
